# Pathogenesis, Immunology and Management of Dermatophytosis

**DOI:** 10.3390/jof8010039

**Published:** 2021-12-31

**Authors:** Shishira R. Jartarkar, Anant Patil, Yaser Goldust, Clay J. Cockerell, Robert A. Schwartz, Stephan Grabbe, Mohamad Goldust

**Affiliations:** 1Department of Dermatology, Vydehi Institute of Medical Sciences and Research Centre University—RGUHS, Bengaluru 560066, India; dr.shishira@gmail.com; 2Department of Pharmacology, Dr. DY Patil Medical College, Navi Mumbai 411018, India; anantd1patil@gmail.com; 3Department of Architecture, Faculty of Art and Architecture, University of Mazandaran, Babolsar 4741613534, Iran; yaser.goldust@gmail.com; 4Departments of Dermatology and Pathology, The University of Texas Southwestern Medical Center, Dallas, TX 75390, USA; ccockerell@dermpath.com; 5Cockerell Dermatopathology, Dallas, TX 75235, USA; 6Dermatology, Rutgers New Jersey Medical School, Newark, NJ 07103, USA; raschwartz@gmail.com; 7Department of Dermatology, University Medical Center Mainz, 55131 Mainz, Germany; stephan.grabbe@unimedizin-mainz.de

**Keywords:** dermatophytes, treatment, pathogenesis

## Abstract

Dermatophytic infections of the skin and appendages are a common occurrence. The pathogenesis involves complex interplay of agent (dermatophytes), host (inherent host defense and host immune response) and the environment. Infection management has become an important public health issue, due to increased incidence of recurrent, recalcitrant or extensive infections. Recent years have seen a significant rise in incidence of chronic infections which have been difficult to treat. In this review, we review the literature on management of dermatophytoses and bridge the gap in therapeutic recommendations.

## 1. Introduction

Dermatophytoses are superficial fungal infections caused by dermatophytes affecting the skin, hair and/or nails [1]. They are also termed as tinea infections. Dermatophytes are filamentous fungi that invade and feed on keratinized tissue like skin, hair and nails, causing an infection [2]. Dermatophytes are divided into nine genera, of which *Trichophyton* (usually affecting skin, hair and nails), *Epidermophyton* (usually affecting skin) and *Microsporum* (usually affecting skin and hair) cause infection in humans. *Trichophyton rubrum* is the most common isolate observed in infections of the feet, body and nails. Depending on the mode of transmission, they are classified as anthropophilic (from humans), zoophilic (from animals) and geophilic (from soil). They are clinically classified by infection site as tinea capitis (head), tinea faciei (face), tinea barbae (beard), tinea corporis (body), tinea manuum (hand), tinea cruris (groin), tinea pedis (foot) and tinea unguium (nails). Other clinical variants include tinea imbricata, pseudoimbricata, and Majocchi granuloma [3]. Recent years have seen an increasing prevalence of dermatophytic infections across the world, especially in tropics. Though not a life-threatening disease alone, it may significantly affect the quality of life [4]. A recent increase in the prevalence, recurrence and resistance could be attributed to the changing epidemiology. In this article, we discuss these trends while providing a brief note on the pathogenesis and diagnosis and management of dermatophytic infection and its relevance in day-to-day practice.

## 2. Changing Trends in Epidemiology

Dermatophytoses, as common superficial fungal infections worldwide, have a higher incidence in tropical and subtropical countries like India due to the presence of high humidity and environmental temperature. Increased urbanization, occlusive footwear, and tight clothing also predispose to higher prevalence [5]. In the last few years, studies have shown a rising trend in the prevalence and also noted a change in the spectrum of infection along with the isolation of previously uncommon species [6,7,8,9]. The rising trend of recalcitrant dermatophytoses could be due to an epidemiological shift in the growth patterns of dermatophytes providing them with advantages of better survival and persistence, an evolution in the genetic make-up of the fungi, enhancing their virulence and pathogenicity, rapid emergence of drug-resistant species due to the rampant use of inadequate doses of potent antifungals. Studies reported a recent shift in the organisms isolated from *T. rubrum* to *T mentagrophytes* complex [8] in Western India and *M audounii* [7]. Recent studies have shown increasing proportion of *T mentagrophytes complex*, mainly *T mentagrophytes var mentagrophytes*. Also, these species showed increased minimum inhibitory concentrations to the commonly used antifungal agents [10].

## 3. Predisposing Factors

The complex interplay between agent, host and the environment plays a role in the pathogenesis of dermatophytoses. The predisposing factors in the host include immunocompromised states such as diabetes mellitus, lymphoma and chronic illnesses, which can lead to extensive, recurrent or recalcitrant dermatophytoses. Intertriginous areas including groin, axilla, inter-web spaces are more susceptible to infection due to excess sweating, rubbing and alkaline pH. Environmental factors which predispose people to higher chances of infection include high humidity, high temperature, increased urbanization, use of tight-fitting clothes and occlusive footwear. In most parts of the world, anthropophilic *T. rubrum* is the most common isolate, but it is being increasingly replaced by *T. interdigitale* and *T. mentagrophytes* complex in some geographical locations [11,12,13]. *T. interdigitale* is responsible for mild and chronic infections [9]. Variations in fungal virulence in various species of dermatophytes are likely to play a role in the recurrence or resistance of infections. A few clinical types like onychomycosis may have a genetic predisposition. Distal subungual onychomycosis may be inherited in an autosomal dominant fashion [14] with incomplete or variable penetrance. Dermatophytic infections is commonly spread in family members, especially in the case of tinea capitis and tinea pedis.

Under favorable conditions, after inoculation into the host skin, epidermal adhesion occurs within an hour, mediated by adhesins present on the fungal cell wall [15,16]. This is followed by penetration mediated by proteases, serine subtilisin and fungalysin, which digests the keratin and also acts as a potent immunogenic stimulus [15]. Upon antigenic exposure, keratinocytes produce a wide range of cytokines—IL- 8, 16, 22, 1beta, TNF alpha, IFN gamma, etc. to destroy the dermatophytes [17]. In addition, the mannans produced by *T rubrum* result in lymphocyte inhibition.

## 4. Immunopathogenesis of Dermatophytoses

There are various host defense mechanisms that avoid establishment of infection, including composition (physical or chemical) of skin, exposure to UV light, lack of humidity, temperature, etc. [16,18].

The several host defense mechanisms against dermatophytoses include increase in cell turnover rate, increase in antimicrobial peptides—beta defensin 2, 3, psoriasin, RNAse7, neutrophil and macrophage-mediated fungal phagocytosis and a complex immune response. The immune responses to dermatophytoses range from an innate immune response to humoral- and cell-mediated immune responses. Cell-mediated immunity plays an important role in the control of dermatophytoses.

### 4.1. Innate Immune Response 

Reduction in the number of epidermal dendritic cells, especially Langerhan cells, in the epidermis, increases the risk of dermatophytosis [19]. These cells contain pattern recognition receptors (PRP) like toll−like receptors (TLR), C-type lectin receptors (CLR) and galectins, which sense the pathogen- associated molecular patterns (PAMPs) on the fungi. Dectins 1 and 2, CLRs expressed in most dendritic cells, recognize the cell wall carbohydrate molecule β-glucan, which activates TLR 2 and 4, causing production of pro-inflammatory cytokines like IL- 6, 10,12, 17 and TNF α, all of which stimulate the adaptive immunity [20,21]. In addition to keratinocytes and dendritic cells, neutrophils also play a pivotal role in innate immunity against dermatophytes. Neutrophils and macrophages are considered to be the final effector cells in mediating extra and intracellular lysis of the fungus via oxidative pathway and through release of TNF α [22,23]. A study demonstrated the role of innate immune response as evidenced by expression of IL-17 as early as 3 days post fungal infection [24].

### 4.2. Acquired Immune Response

The comparative role of humoral and cell mediated immunity against dermatophytes has been debated. While cell-mediated immunity is protective against fungi, certain antibody responses may also provide protection. Cell-mediated immunity increases the epidermal proliferation, thus increasing cell turnover and facilitating the elimination of fungi [11,25]. Overall, elimination of dermatophytoses is mediated by Th 1 type of cell-mediated immunity while Th2 response predisposes to infection or results in an allergic response. Th1 cells produce cytokines like IFN α and stimulate phagocytosis [26,27]. Th2 response results in production of immunoglobulins and IL- 4, 5, 13 [24,28]. Initially IFN γ levels are low and IL-10 is high, inhibiting the Th1 response. Over time, there is decrease in IL-10 and a rise in IFN γ. Thus, IL-10 induces a Th2 response and also plays a significant role in innate immunity and immune response regulation. IL-10 also prevents TNF α production, enabling development of a specific immune response [29]. Recently, the importance and protective role of Th17 cells promoting the Th1 immune response and inhibiting the Th2 response has been discussed. IL-17 α aids in mobilization of neutrophils and stimulates defensins, thus resulting in rapid effective control of infections [30,31].

Acute dermatophytosis is associated with a delayed type hypersensitivity reaction, while persistent infections correlate with inadequate cellular immune responses with immediate hypersensitivity responses, high levels of IgG4 and IgE antibodies and release of Th2 cytokines [32,33]. Similarly, chronic dermatophytosis due to association of immediate hypersensitivity and Th2 cytokines, may have an underlying pathogenesis of allergic diseases mainly asthma [20].

## 5. Management 

### Laboratory Diagnosis

The evolution of clinical presentation and varied manifestation poses a practical difficulty in differentiating dermatophytoses from non-dermatophytic or non-mycotic dermatoses. Hence, to initiate appropriate therapy, there is a need for appropriate laboratory diagnoses. To obtain optimal results, the quality and quantity of the clinical material examined is critical. Skin scrapings should be collected from the edge of the lesion and transported on a sterile black chart paper to keep the sample dry, thus preventing bacterial contamination. This practice is in accordance with the expert consensus [34] and also reappraised by Pihet [35] et al. in a recent review. The skin scrapings can be collected using scalpel blades, curettes or the edge of a slide [36,37]. In onychomycosis, the nail clippings should be collected as proximal as possible due to the fact that there are more hyphae and more viable hyphae in the proximal part of the nail.

Direct microscopic examination—10–40% KOH mount (10% for skin scrapings and 40% nail clippings) is useful for rapid detection of dermatophytoses. The procedure is simple, inexpensive, rapid, and efficient screening method as highlighted by Kurade et al. [36], Pihet et al. [35] and Mc Kay et al. [37]. The presence of long, smooth, refractile, branching, undulating, septate hyphal filaments with or without arthroconidiospores, indicates positive scrapings. The adequacy of sample, the appropriateness of the tool used for sample collection and expertise of the physician determines the sensitivity and specificity of the diagnosis [34]. Addition of 36% dimethyl sulfoxide (DMSO) [38] to KOH solution shortens the time necessary for clearing, does not dry out the fluid, accelerates clearing of thick scales and imparts transparency to the keratinocytes, aiding in better visualization of the fungal elements. An alternate contrast stain, Chicago Sky Blue stain [38] is used, along with 10% KOH as a clearing agent. This stains the fungal hyphae blue against a pink background making it easier to identify them. Fluorescent staining of the specimen with an optical brighteners (diaminostilbene), which binds to chitin, a cell wall component of the fungi, is the most sensitive method to microscopically detect fungi in skin, hair and nails [39]. Flurochromes [40] like Calcofluor or Blankophor in combination with fluorescent microscopy makes identification easier, faster and safer than KOH examination. These fluorochromes binds non-specifically to chitin and glucan which are components of the fungal cell wall. On exposure to ultraviolet light during fluorescent microscopy, the fungal filaments and spores appear blue-white in color. Chlorazol black E is a stain having high affinity for chitin, found in fungal cell walls but not in vertebrate tissues. It stains the cell walls of filamentous fungi and yeast, a blue-black color. Chorazol black E-stained wet mounts of fingernail and toenail samples are a valuable diagnostic method for onychomysosis, because it accentuates the presence of even small numbers of fungal hyphae. It has a high specificity of 98%, and hence can be used as a confirmatory test [41]. Lactophenol Cotton Blue mount [42], aids in microscopic examination of fungal growth characteristics like the nature of the mycelium, presence or absence of microconidia, shape and morphology of macroconidia, which helps in identification of various species of dermatophytes.

Fungal culture aids in definitive identification of fungal species. Though it is considered the gold standard in the diagnosis, its routine use is often omitted as it lacks sensitivity, has a prolonged turnaround tine and is not readily available [35]. However, its use is recommended in special situations like recalcitrant or extensive cases. Sabaurouds dextrose agar (SDA) is the most commonly used medium. Modified SDA with additional gentamicin, chloramphenicol and cycloheximide is more selective for dermatophytes [43]. Dermatophyte test medium (DTM) is another isolation medium containing a pH indicator—phenol red. After incubation at room temperature for 5–14 days, the color of the media changes from yellow to bright red as the dermatophytes utilize proteins resulting in ammonium ion release and an alkaline environment. However, in nail infections, dermatophyte test medium is not routinely used because of too many false positives caused by rapid growers and other contaminants. Potato dextrose agar (PDA) contains dextrose as a carbohydrate source, a growth stimulant and potato infusion, nutrient base for luxuriant growth of fungi. Addition of a specified amount of tartaric acid (10%), inhibits bacterial growth by reducing the pH of the medium. 

Dermatophyte antifungal sensitivity testing has been well recognized by experts [34]. However, current routine use is not feasible in real time settings as it lacks a consistent correlation between the in-vitro data and clinical outcome.

Species identification is based on the characteristics of colony, morphology and physiologic tests. Identification of dermatophytes can be done based on their macro- or microconidia features. 

Dermoscopy has an adjuvant role in the identification of dermatophytosis. Therapeutic implication of involvement of vellus hair in tinea corporis is an indicator to start systemic therapy [3]. In tinea capitis, the presence of comma hair, corkscrew hair and fractured hair shafts have been seen [44]. Dermoscopy is useful in diagnosis of onychomycosis [45]. In distal lateral subungual onychomycosis (DSO), the proximal edge of the onycholytic area is jagged, with indentations caused by sharp structures called spikes and detached nail plates have an irregular matte pigmentation distributed in striae, giving an overall aurora borealis-like appearance. These three dermoscopic diagnoses have 100% specificity for DSO.

Histopathology has been used in Majocchi granuloma. Hyphae can be appreciated in the stratum corneum on H&E staining. Periodic acid Schiff staining and Gomori methanamine silver stains help in highlighting the fungal hyphae.

Onychomycosis severity index [46], is a simple tool to grade the percentage involvement of nail plate, proximity of infection to nail matrix, presence of dermatophytoma and degree of subungual hyperkeratosis. Nails with low score are more likely to respond favourably to conventional therapy, while nails with high scores would be more difficult to treat.

Molecular methods [47] have been developed to provide more rapid and accurate alternatives to existing dermatophyte identification methods due to overlapping phenotypic characteristics, variability and pleomorphism. These include gene-specific PCR, sequencing of r-RNA gene, chitin synthase encoding gene, PCR fingerprinting and DNA hybridization. Sequencing of internal transcribed spaces (ITS) regions has proved to be a useful method for the phylogenetic analysis and identification of dermatophytes. Though useful, the main hindrance to its application in routine practice is the high cost associated with the sequencing.

Polymerase chain reaction (PCR) and nucleic acid sequence-based amplification these not only help in rapid diagnosis but also in detecting drug resistance [3]. Uniplex PCR is useful for direct fungal detection in clinical samples with sensitivity and specificity of 80.1% and 80.6% respectively in comparison to culture [48]. Multiplex PCR for fungal detection in dermatophytes enables detection of 21 dermatomycotic pathogens with DNA detection using agarose gel electrophoresis.

Matrix assisted laser desorption ionization- time of flight mass spectrometry (MALDI-TOF-MS) is based on detection of biochemical characteristics, proteolytic degradation product as a result of fungal activity [34]. It is a promising experimental technique, highlighted as a first line, economical, accurate and faster identification method. However, it cannot be considered a practical tool as a pre-requisite of culture is mandatory and it cannot be done on clinical samples and is not readily available.

Reflectance confocal microscopy, a new diagnostic technique that provides in-vivo imaging of epidermis and superficial dermis at cellular level to detect fungal infections and parasitic infestations on skin [49]. It is noninvasive and has shown to have 100% sensitivity in a retrospective analysis [49].

## 6. Treatment

An ideal treatment should have a high cure rate with a low relapse rate, strong anti-inflammatory action, rapid onset of action, short duration of action with no side effects, minimal systemic absorption and should be cost effective, safe to be used in pregnancy ad lactation and in renal and hepatic failure.

### 6.1. General Measures

Use of loose-fitting clothes made of cotton should be encouraged. Emphasis should be given on importance of regular use of medicines. Sharing of bed linens, towels, clothes and shoes should be avoided. All the clothing, especially socks, caps, undergarments, should be washed in boiling water, sun dried and ironed before reuse. Patients with risk factors like obesity or excessive sweating should be encouraged to use absorbent powders, deodorants (to reduce perspiration), frequent changes of clothes and advised weight reduction appropriately. In tinea pedis, prophylactic use of anti-fungal powders is advised. Patients should avoid the use of occlusive footwear.

### 6.2. Medical Management

Topical anti-fungal therapy forms the mainstay of treatment in localized and naive dermatophytic infections. Topical anti-fungal agents are recommended for dermatophytoses affecting skin and presenting as localized infections. Systemic drugs are indicated for more extensive infections. Table 1 summarizes the classification of topical and systemic anti-fungals [50,51]. Combination therapy is expected to have a better clinical and mycological cure than topical or systemic agents used alone. It is preferred to use combination from different group of antifungals not only for wide coverage, but also for preventing emergence of resistance.

Indication for use of systemic antifungals [3] (Table 2, Table 3 and Table 4)

Tinea capitisTinea unguiumDermatophytic infection involving more than one region simultaneously—tinea corporis and cruris, tinea cruris and pedisExtensive tinea corporis. However, there is no standardized definition of extensive infectiondExtensive tinea pedis involving the sole, heel and dorsum of the footResistant or recalcitrant or chronic dermatophytosis or patients who fail with topical therapy.

Various topical agents are available for use in dermatophytoses (Table 5). Various traditional drugs with no specific antifungal activity [3] such as Whitfields’s ointment and Castellani’s paint are still in use, however, their efficacy has not been well quantified [3]. A meta-analysis by Rotta et al. [52] evaluated the efficacy of antifungals involving 14 topical agents. It included 65 randomized controlled trials (RCT) comparing the efficacy of topical agents with one another and placebo. Mycological cure and sustained cure were considered for evaluation of efficacy. There was no statistically significant difference among antifungals with respect to mycological cure. For sustained cures, butenafine and terbinafine was found to be superior to clotrimazole. In pairwise comparison butenafine and terbinafine was found to be superior to oxiconazole, clotrimazole and sertaconazole. Terbinafine was superior to ciclopirox and naftifine superior to oxiconazole [3]. A Cochrane Review on use of topical antifungals in tinea corporis and cruris reported terbinafine and naftifine to be effective with fewer side effects [53]. Topical azoles were also found to be effective as estimated by clinical and mycological cure. Difference in efficacy between antifungals is mainly due to fewer applications and shorter duration of treatment compared to others. As per the Cochrane Review, systemic therapy is used for mainly chronic infection or failure of topical therapy and in severe forms of the disease. In a meta-analysis of 11 RCTs terbinafine and naftifine showed slightly higher cure rates than an azole [54]. Nystatin is not effective in dermatophytic infection, while topical naftifine gel is effective in both interdigital and moccasin type of tinea pedis [55].

#### 6.2.1. Tinea Corporis/Cruris

Expert consensus recommends the use of topical antifungal in naive cases and when the infection is limited to superficial keratinized tissue [34]. Superiority of any given class of topical antifungal has not been established clearly in clinical trials [34]. However, as per the current situation, experts favor the use of azoles over allylamines due to their antibacterial, anti-inflammatory and broad-spectrum antifungal properties [56]. Systemic therapy is indicated in extensive lesions or lesions with papulopustules, dermoscopic involvement of vellous hair and recalcitrant infection. Terbinafine (250 mg) once daily and itraconazole (100–200 mg/day) are equally effective in naive cases. In recalcitrant cases, itraconazole (200–400 mg) in divided doses along with topical antifungals is recommended by the experts [3]. Minimum duration of therapy should be 2–4 weeks in naive and 4 weeks in recalcitrant cases. The value of examining and identifying close contacts and their treatment should be emphasized. Patients should be advised to avoid body contact sports.

#### 6.2.2. Tinea Incognito

This is a superficial dermatophytic infection modified by improper use of steroids or utilization of immunomodulators, making it difficult to diagnose clinically [57]. The classical picture is masked and suppressed but the lesions relapse on stopping the administration of creams. Also, it can present as purulent folliculitis. Expert consensus recommends abrupt stopping of topical corticosteroids, except for steroid-induced rosacea. Itraconazole (200–400 mg per day) for 4–6 weeks is recommended in these patients [34].

Systemic antifungals are indicated in extensive involvement and those for whom topical treatments fail. Terbinafine and itraconazole are commonly prescribed. Griseofulvin and fluconazole are effective, but require prolonged duration of the treatment. A study comparing itraconazole (100 mg/day) with ultramicronized griseofulvin (500 mg/day) for tinea corporis/cruris showed significantly higher clinical and mycological outcome with itraconazole after 2 weeks [58]. A similar study comparing griseofulvin and terbinafine (both 500 mg/day for 6 weeks), showed mycological cure rates of 73% and 87%, respectively [59]. Another double blind study, comparing itraconazole (100 mg/day) and griseofulvin (500 mg/day), found the former to have better efficacy in providing clinical and mycological cures [60]. A systematic review of systemic antifungals showed terbinafine to be more effective compared to griseofulvin. Terbinafine was found to have similar efficacy to itraconazole [61].

#### 6.2.3. Tinea Pedis

Fungal infections of the foot are classified based on the site i.e., sole of the foot, lateral aspect of foot (moccasin) (most severe and chronic form) and in between the toes (interdigital; the most common type). Tinea pedis may be associated with secondary bacterial infections, immunological reactions and perifollicular granulomatous inflammation [62].

Topical therapy is the first line treatment. Since, secondary bacterial infections can occur, so it is recommended to use antifungals with antibacterial properties like ciclopiroxolamine, miconazole nitrate, naftifine hydrochloride and sulconazole. Patient compliance is very important, as patients tend to stop therapy on improvement of clinical symptoms, prior to completing the full regimen, thus contributing to reinfection or relapse. 

Systemic therapy is given in severe cases and in those of non-response to topical therapy. Terbinafine (250 mg/day) for 2–6 weeks has clinical cure rates of 75–85% and mycological cure rates of 77–83% at the end of 6 weeks [63]. Itraconazole (200 mg twice daily) for 7 days showed a mycological and clinical cure rate of 79% and 93%, respectively [64]. Fluconazole (50 mg daily or 150 mg/week for 6 weeks) had a mycological and clinical cure rate of 93% and 79%, respectively.

Various topical agents, including terbinafine, butenafine, naftifine, ciclopirox, clotrimazole, econazole, miconazole, sertraconazole and luliconazole, have been used. Mycological and clinical cure was achieved in 67–95% and 60–80% of cases, respectively, with 1% terbinafine cream [65]. With terbinafine spray mycological and clinical cure rates range from 85–92% and 66–83%, respectively. Ciclopiroxolamine is FDA approved for use in tinea pedis caused by *T. rubrum*, *T. interdigitale*, *E. floccosum* and *M. canis* [66]. When used as 0.77% cream/gel twice daily for 4 weeks it resulted in clinical and mycological cure rates of 51.4% and 88.8%, respectively, after 8 weeks. Side effects were comparable to placebo [67]. Sertaconazole 2% cream twice daily for 4 weeks produced a high efficacy rate of 88.9–90.6% with no side effects. Luliconazole 1% cream once a day for 2 weeks showed complete clearance in 26.4% in 28 days post-treatment. Econazole 1% foam has been recently approved by the FDA. This novel water lipid delivery system aids in drug delivery and restoration of damaged skin [68] Foam formulations are easier to apply and aesthetically more pleasing. Once a day application showed a mycological cure of 67.6% after 4 weeks. Trials have shown the efficacy of naftifine gel in treating moccasin tinea pedis. Naftifine has antifungal, anti-inflammatory and antibacterial properties [69]. Phase III trials with gel used once daily for 2 weeks showed complete cure in 19.2% of patients and a mycological cure rate of 65.8% at 4 weeks post-treatment [55]. 

#### 6.2.4. Tinea Capitis

This is a superficial fungal infection located primarily in the hair follicles and surrounding skin. The goals of treatment include alleviation of symptoms, clinical and mycological cure and prevention of further transmission. Systemic therapy is indicated to achieve mycological cure. Griseofulvin is FDA approved for treatment and has 88–100% efficacy. Terbinafine is FDA-approved for use in children >2 years of age; treatment is usually prolonged for 6–8 weeks. Efficacy rates of terbinafine and griseofulvin are dependent on the causative organism. T capitis due to *T tonsurans* responded better to terbinafine (47.7–56.1%) than griseofulvin (34.4–36.5%). Infections caused by *M. canis* responded better to griseofulvin (35.1–51.1%) compared to terbinafine (23–30.6%) [70,71]. Itraconazole pulse therapy is effective and improves patient compliance in children. It is given at a dose of 5 mg/kg/day for a week, with a 2 week gap between the second and third pulse; thus produced complete clearance of 100% 12 weeks after treatment [72]. Shampoos like selenium sulfide, ciclopirox and zinc pyrithrone twice weekly, given in combination with systemic therapy, are a preventive measure to treat asymptomatic carriers [63].

#### 6.2.5. Tinea Unguium

This is dermatophytic infection of the nail plate with high recurrence rates, which is difficult to treat, affecting 12–13% of the population [66]. Most common species causing nail infections are *T. rubrum* and *T. interdigitale*. The most common presentation is distal lateral subungual onychomycosis. The success of treatment depends on the nail growth. Fingernails show faster growth than toenails, hence need shorter treatment and have higher success rates compared to toenails, which usually take 12–18 months to grow [63,73]. The gold standard of cure in Tinea ungiuim is clinical cure, not mycological cure. Complete cure is defined as combination of both clinical and mycological cure. A combination of topical and systemic therapy is recommended. Nail avulsion with urea followed by bifonazole is a treatment option and is more attractive than surgical nail avulsion. Commonly used drugs are terbinafine and itraconazole and fluconazole, with VT 1161, in early development. Systemic antifungals like terbinafine and itraconazole are prescribed for 12 weeks for toenails and fluconazole for 9–18 months. Topical antifungals need to be continued for up to 48 weeks until healthy nail growth is complete.

Topical antifungals are mainly used for mild to moderate cases of onychomycosis or for single nail infections. Amorolfine 5% and ciclopirox 8% are the most commonly used agents, while efinaconazole 10% and tavaborole 5% have been recently introduced [66]. Also, laundering gloves and socks in 60^◦^ C wash for 45 mins aids in destroying residual fungal matter [74]. Maintenance of proper nail and foot hygiene, avoiding occlusive footwear and ill-fitting shoes, timely treatment of cases and contacts may also decrease the risk of recurrence or reinfections [75].

Systemic drugs can be given as daily or pulse therapy. A meta-analysis concluded that 250 mg daily terbinafine and 400 mg pulsed itraconazole were superior to fluconazole and other topical agents [76]. Though multiple regimens are available for terbinafine, a meta-analysis supports continuous terbinafine over pulse regimen [77]. Itraconazole is found to be effective as a continuous regimen of 200 mg/day for 12 weeks or as pulse therapy of 200 mg twice daily for a week, then a 3 week gap. Such three pulse treatments are recommended for toenails and two pulses for fingernail infections. Itraconazole has advantage that it is not only used for tinea unguium, but also effective for mixed and non- dermatophytic mold infections. 

## 7. Special Situations

### 7.1. Majocchi Granuloma

This is a deep dermatophytic infection that occurs due to progressive dissemination of the fungus into the subcutis, most commonly caused by *T rubrum*. Any trauma can allow fungi to reach the reticular dermis leading to cellular destruction reducing the dermal pH, hence, making the milieu suitable for fungal survival, most commonly seen in immunocompromised individuals. Inadvertent use of topical steroids is a risk for its development. Systemic antifungals like terbinafine 250 mg/day for 4–6 weeks, itraconazole 400 mg/day for a week every month for 2 months, have shown to be successful treatment options [3].

### 7.2. Immunosuppression and Pregnancy

In patients with HIV, dermatophytosis tends to have more extensive and characteristic morphology that may not be seen due to their suppressed immunity. In a patient with chronic illness like renal/hepatic impairment or patients on polypharmacy, caution should be taken while prescribing systemic antifungals. In patients with renal impairment, terbinafine clearance is reduced significantly, hence, dose adjustment is advisable or alternate drug should be preferred. In hepatic impairment, itraconazole should be avoided [3,34].

In pregnancy, systemic antifungal use should be avoided. Category B topical drugs such as clotrimazole, terbinafine, ciclopirox, naftifine and oxiconazole are preferred. Econazole, miconazole, ketoconazole, selenium sulfide are category C drugs and should be avoided. Some of these oral medications are high risk for the extremely rare but potentially life-threatening drug reaction called Steven-Johnson syndrome/toxic epidermal necrosis (Ref). Terbinafine is the only oral antifungal with pregnancy category B. However, adequate data on its use is not available [3,34].

### 7.3. Elderly

Treatment should be tailored as per patient need, site and extent of involvement. Comorbidities and possibility of drug interactions should be taken into account prior to starting therapy [3,34].

### 7.4. Children

This infection is relatively less common, with 3% prevalence in India. We recommend the use of topical agents, owing to rapid cell turnover in children, which facilitates better clinical outcomes [34]. In recalcitrant/extensive dermatophytosis, fluconazole or terbinafine (>2 yrs) can be used with caution [34].

Factors Responsible for Clinical Failure of Antifungal Therapy [78].

Three factors play a role in the clinical failure of therapy—recurrence, resistance and remedy. Recurrence—the clinical failure to antifungals—is either due to persistent or recurrence of infection, presenting clinically as an initial response followed by extension/spread of lesions or as complete clinical remission followed immediately by reappearance of lesions or no clinical response to antifungals. Such situations indicate recalcitrant clinical types, persisting predisposing factors and drug resistance. Recalcitrant clinical types—chronicity—can result in extension and spread of the lesions due to inadequate treatment. Also, they act as a reservoir of infection. Involvement of vellus hair can lead to clinical failure of topical therapy and warrants the use of systemic antifungal agents. Also, invasive dermatophytosis and tinea unguium act as a reservoir of infection and can also result in chronic infection. Persisting predisposing factors—various host and environmental factors—promote recurrence or persistence of infection like presence of comorbidities, changes in climate, lifestyle, attitude towards health. Hot and humid climate, tight fitting clothes provide occlusive milieu where dermatophytes can thrive. Attitudinal changes such as reluctance to seek medical opinion, non-compliance, unrealistic demand of quick relief and self-medication augment the existing problem. Inappropriate antifungal therapy—inappropriate selection, inadequate dose and duration of therapy result in partial response or rapid recurrence of infection, also facilitates drug resistance. Self-medications with topical steroid- containing creams increase the prevalence of recalcitrant clinical variants. Quality of drugs—due to increased availability of various brands of antifungals and issues related to quality control are a real concern. Use of less efficacious molecules can lead to clinical failure and drug resistance. Resistance—the selective pressure of immune response and antifungal agents—results in selection of resistant strains. The mechanisms of drug resistance include decrease in drug uptake, structural alterations in target site, increase in intracellular targets, increased drug efflux and formation of biofilms. The in vitro resistance is determined by MIC, but in dermatophytosis the in vitro resistance based on MIC does not always correlate with clinical resistance. Hence, MIC of an antifungal drug is not the sole factor predicting clinical cure or clinical failure. Remedy—modification of pharmacotherapy—is not the only solution. There is a need to formulate an effective strategy to literate the population regarding predisposing factors, adverse effects of over-the-counter drugs, need to expert opinion, importance of adhering to expert’s advice and preventive measures to control drug resistance by judicious use of antifungal agents, in proper doses and preferring use of combination therapy with different mechanism of action. Also, new targets have been identified for new antifungal agents to combat drug resistant strains. The putative targets include genes and proteins, virulence factors like keratinase, proteases, elastases, lipase, sulfite transporters, heat shock protein 90 and ATP binding cassette transporters.

## 8. Conclusions

Successful management of dermatophytosis has increasingly become challenging owing to the changing epidemiological factors and the emergence of drug resistant organisms. Appropriate dose and duration of drug in a compliant patient helps achieve successful mycological cure. In addition to pharmacological therapy, general measures and lifestyle changes also play a crucial role in preventing recurrences. Improved diagnostic tests and novel immunomodulatory therapy portend advances in disease management.

## Figures and Tables

**Table 1 jof-08-00039-t001:** Classification of systemic antifungals.

Class	Representative Drugs
Heterocyclic benzofuran	Griseofulvin
Azoles	
Imidazoles	Topical clotrimazole, econazole, miconazole, oxiconazole, luliconazole, butoconazole, fenticonazole systemic ketoconazole
Triazoles	Fluconazole, itraconazole, voriconazole, posaconazole, isavuconazole
Allylamines	Naftifine, terbinafine, butenafine
Benzylamines	Butenafine
Echinocandins	Caspofungin, micafungin
Piridone derivatives	Ciclopirox olamine
Antimetabolite	Flucytosine
Oxaborole	Tavaborole
Thiocarbamate	Tolfanate
Morpholine derivatives	Amorolfine HCl
Others	Undecylenic acid, Whitfield ointment, BPO, zinc pyrithione, selenium sulphide, azelaic acid, nikkomycin, icofungipen, triclosan, eucalyptus oil, dermcidin, macrocarpal C, tetrandrine

**Table 2 jof-08-00039-t002:** Dose and duration of systemic antifungals in dermatophytic infection.

	Fluconazole	Griseofulvin	Itraconazole	Terbinafine
Tinea Capitis	6 mg/kg/day × 3–6 weeks	10–15 mg/kg/day (ultramicrosize) 20–25 mg/kg/day (microsize suspension) × 6–8 weeks	5 mg/kg/day × 4–8 weeks	Adults:250 mg/day × 3–4 weeks. Children:Granules: 125 mg (<25 kg), 187.5 mg (25–35 kg) or250 mg (>35 kg) × 3–4 weeks
Tinea Corporis/Cruris	2–4 weeks	2–4 weeks	1 week	1 week
Tinea Unguium	3–4 months for fingernails. 5–7 months for toenails	1–2 g/day (microsize) or 750 mg/day (ultramicrosize) until nails are normal	200 mg/day × 12 weeks or 200 mg twice a day (BID) × 1 week/month for 2–4 consecutive months	12 weeks 6 weeks
Tinea pedis	4–6 weeks	4 weeks	1 week	2 weeks

**Table 3 jof-08-00039-t003:** Paediatric and adult dosing of systemic antifungals.

Systemic Antifungal	Per kg Body Weight Dose	Adult Dose
Fluconazole	6 mg/kg/week	150–450 mg/week
Griseofulvin	15–20 mg/kg/day (microsize suspension) 10–15 mg/kg/day (ultramicrosize suspension)	500 mg/day
Itraconazole	3–5 mg/kg/day	200 mg/day
Terbinafine		250 mg/day

**Table 4 jof-08-00039-t004:** Mechanism of action of systemic antifungal agents.

Drug	Mechanism of Action
caspofungin	Fungal cell wall synthesis inhibition
Amphotericin-B, Nystatin	Binds to fungal cell membrane ergosterol
Terbinafine	Inhibition of lanosterol and ergosterol synthesis
Azoles	Inhibition of ergosterol synthesis
5-Flucytosine	Inhibition of nucleic acid synthesis
Griseofulvin	Disruption of mitotic spindle and inhibition of fungal mitosis

**Table 5 jof-08-00039-t005:** Summary of topical antifungals in dermatophytic infections.

Azole	Preparations	Site	Frequency of Application	Duration of Use
Imidazoles (%)				
Clotrimazole (1)	Cream, lotion	T. corporis/cruris/pedis/capitis	BD	4–6 weeks
Econazole (1)	Cream	T. corporis/cruris/pedis/capitis	OD-BD	4–6 weeks
Miconazole (1)	Cream, lotion	T. corporis/cruris/pedis/capitis	BD	4–6 weeks
Oxiconazole (2)	Cream, lotion	T. corporis/cruris/pedis/capitis	OD-BD	4 weeks
Sertaconazole (2)	Cream	T. corporis/cruris/pedis/capitis	BD	4 weeks
Luliconazole (1)	Cream, lotion	T. corporis/cruris/pedis/capitis	OD	2 weeks
Eberconazole (1)	Cream	T. corporis/cruris/pedis/capitis	OD	2–4 weeks
Triazoles (%)				
Efinaconazole (10)	Solution	T. pedis	OD	Up to 52 weeks in co-existing tinea unguium
Allylamines				
Terbinafine	Cream, powder	T. corporis/capitis	BD	2 weeks
		T. cruris	BD	2 weeks
		T. pedis	BD	4 weeks
		T. manum	BD	4 weeks
Naftifine 1%	Cream	T. corporis/cruris/pedis/capitis	OD-BD	Use 2 weeks beyond resolution of symptoms
Butenafine 1%	Cream	T. corporis/cruris/pedis	OD-BD	2–4 weeks
Others				
Amolorfine 0.25%	Cream	T. corporis	BD	4 weeks
Amphotericin B (1 mg) 0.1%	Lipid-based gel	T. corporis	BD	2 weeks
Ciclopiroxolamine 1%	Cream, lotion	T.corporis/cruris/pedis	BD	2–4 weeks

## Data Availability

Not applicable.

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
