# Peer review of "Pathogenesis, Immunology and Management of Dermatophytosis"

_jof, 2021, doi:10.3390/jof8010039_

Round 1

Reviewer 1 Report

A topical work summarizing and updating the latest views on the epidemiology of fungal infections. The topic is very important for practicing physicians and dermatologists. Thorough review of available literature, current data. I recommend the publication

In my opinion, it fits the theme of the special issue and is valuable and practical.

This work contains current evidence and data on the epidemiology of mycoses and the variability of the causative species. It is observed that the dominant strains change due to migration, climate changes and implemented treatment. The analysis of mutual mechanisms of pathogen-host interactions involving both innate and acquired immune mechanisms seems to be particularly valuable. Such works are increasingly appearing in various skin infections accompanying systemic diseases. The analysis of the treatment of particular types of mycoses, distinguishing between drugs and dosage regimens, seems to be particularly important in practice. This makes the reading easier and can be extremely helpful. 

Author Response

Thank you for your valuable review and comments.

Reviewer 2 Report

pag 46 line 1. tinea infecton CHAGE dermatophitic infection

pag 3 lines 106 Trichophyton ITALICS

LABORATORY directo microcopic examination ADD chlorazol black

pag 8 line 283 T. tonsurans ITALICS, lines 320 T. rubrum  ITALICS 

Author Response

Many thanks for your nice comments dear reviewer.

All the corrections have been incorporated.

The changes have been highlighted in red in the document.

Tinea infection changed to dermatophytic infection – pg 6 line 265

Trichophyton has been written in italics – pg 3, line 121

In laboratory diagnosis – we have incorporated chlorazol black – pg 3, line 146-151

T tonsurans and t rubrum written in italics – pg 10, line 402, pg 11, line 443

Thank you

Reviewer 3 Report

Introduction:

lines 39-40-Epidermophyton and Microsporum "Usually" affecting. These 2 may rarely affect nails

Line 39-40 Though not a life threatening disease "alone". Dermatophyte infection of the feet and lower legs in the elderly, diabetic or otherwise immunosuppressed, can cause a break in the skin barrier and let staph and strep in that may cause cellulitis. 

Pathogenesis-At least with tinea pedis and onychomycosis, these may be predisposed to in families. Distal subungual onychomycosis may be inherited in an autosomal dominate fashion with incomplete or variable penetrance. Tinea is commonly spread in families , especially tinea capitis and tinea pedis. 

Lab diagnosis: In nail onychomycosis, obtain the specimen as proximal as possible due to the fact that there are more hyphae  and more viable hyphae the more proximal part of the involvement.

line 134-DTM is not good for diagnosing nail dermatophytosis because of too many false positives by rapid growers and other contaminates

line 143-Dermoscopy is helpful in onychomycosis-Dr Piriccini 

Table 2 Fingernail treatment is not  longer than toenail treatment. Check references for the correct treatment duration here.

The gold standard for telling onychomycosis cure is clinical cure, not mycological cure. Complete cure is defined as clinical and mycological cure combined.

Incorporate the following paper: Onychomycosis Severity Index by C. Carney, et. al. Archives of Dermatol. November 2011. 

Author Response

Many thanks for your kind review and nice comments.

All the changes have been incorporated and highlighted in red.

In introduction

Line 39-40 – we have made the changes and added usually affecting – pg1, line 33-34

Line 39-40 – have included alone – pg 1, line 42

Pathogenesis -have incorporated the lines as suggested by the reviewer with appropriate references – pg 2, line 68-72

Lab diagnosis – we have added the lines on onychomycosis – pg 3, line 135-137

Line 134  - DTM limitation has been added as suggested by the reviewer – pg 3, line 160-162

Line 143 – added dr piriccini  - pg 4, line 164-169

Table 2 – corrections made – in table 2

We have incorporated the article as suggested by the reviewer – pg 4, line 172-175

Thank you

Reviewer 4 Report

I approached the review with great interest because this topic is of my personal scientific interest. Nevertheless, the approach presented by the authors is very general and choathic. Perhaps aim set too broadly makes it impossible to discuss the topic in a relatively short article. In addition, most of the references cited are obsolete articles, which does not give the full picture of the problem because knowledge on this topic is updated very quickly. 

Author Response

Many thanks for your precious time and kind review.

Actually, we have added few more articles and have updated the article with the information from the latest publications.

Thank you

Round 2

Reviewer 3 Report

Introduction lines 33-34

delete nail involvement for Epidermophyton

Author Response

Respected reviewer 2

The corrections as recommended by you have been incorporated.

Thank you

Reviewer 4 Report

Despite the attempts to make corrections to the article, I still find that the manuscript is fragmentary, it lacks specific knowledge in the field of dermatophytosis and contains numerous factual errors. Some specific comments below:

line 33-34: Trichophyton, Epidermophyton and Microsporum in italics. Please include this comment throughout the manuscript as there are such errors in many places

line 52-53: The mere indication of a change in the prevalence of isolation of individual species of dermatophytes is insufficient, I propose to include an explanation here. In addition, the information is given in a very general manner, hence incorrect because for many geographic regions, completely opposite changes take place

line 55: This entire section contains only the indicated predisposing factors and has nothing to do with pathogenesis. It is necessary to change the title or expand the information

line 115-116: Does the information provided refer only to dermatophytes classified as Trichophyton?

line 133-145: This entire paragraph is some selected issue according to unknown key. In the preparations, first of all, arthrospores are searched for, which indicates an active infection, and secondly, hyphae, rather as part of a fungus that does not multiply. The authors do not give a word about the addition of DMSO to KOH and this is now even a standard. Finally, the fluorescent dyes and the interpretation of the results in this method are not mentioned, and it is not known why chlorazole black is described with the exclusion of lacotophenol blue. The latter is now routinely used in diagnostics.

line 146-156: To mention only the SDA in the description of the breeding study is at least an oversight. DTM, PDA and a few more additional substrates are selected more often. Why? Due to the fact that on SDA, dermatophytes produce conidia only sporadically and their appearance is the basis for identification using these methods. So what use of such a solution, none

line 156-190: The authors did not even mention ITS analysis later in the chapter. After all, this is now the basis for identification and the entire taxonyma of this group is based on the ITS sequence analysis. Such a presentation of diagnostics is unacceptable and misleads the reader. Maybe in the 1980s it could be considered valuable

Table 1: Mentioning polyenes for the treatment of dermatophytosis is a misunderstanding. So far, these drugs have not been used in the treatment of this disease and this is unlikely to change. I categorically consider it a major factual error

line 193: There is not a word in the entire chapter on clinical resistance, recurrent and chronic cases, and these are the things that should be occupied in the literature. What the authors present is textbook information, nothing scientific. 

Author Response

Thank you very much for the valuable suggestions and opinion. The corrections have been made as per your suggestions and have been highlighted in red.

The corrections are as follows –

Line 33, 34 – words changed to italics

Line 52 – the geographic area has been added and also a note on the changing species has been added.

Line 62 – heading changed to predisposing factors

Line 124 – sir/ Madam, it is for all dermatophytes

Line 142-166 – DMSO, fluorescent staining, LPCB have been added

Line 177- PDA, DTM added

Line 200 – molecular diagnostics methods including ITS sequencing has been incorporated in the article.

Table 1 – polyenes removed from the table

Line 480 – a note has been added on recalcitrant, recurrent and resistant infections

Thank you for your valuable inputs.